# A Novel Approach for Satellite-Based Turbulence Nowcasting for Aviation

**Axel Barleben *** , **Stéphane Haussler, Richard Müller** and **Matthias Jerg**

German Weather Service, Frankfurter Str. 135, 63067 Offenbach, Germany; stephane.haussler@dwd.de (S.H.); richard.mueller@dwd.de (R.M.); matthias.jerg@dwd.de (M.J.)

* Correspondence: Axel.Barleben@dwd.de

**Abstract:** The predictability of aviation turbulence is influenced by energy-intensive flow patterns that are significantly smaller than the horizontal grid scale of current numerical weather prediction (NWP) models. The parameterization of these subgrid scale (SGS) processes is possible by means of an additional prognostic equation for the temporal change of turbulence kinetic energy (TKE), whereby scale transfer terms are used. This turbulence scheme has been applied operationally for 5 years in the NWP model ICON (Icosahedral Nonhydrostatic). The most important of the source terms parameterizes the Kelvin–Helmholtz instability, better known as clear air turbulence. This shear term was subjected to a nowcasting technique, is calculated with satellite data, and shifted forward in time using motion based on optical flow estimates and atmospheric motion vector (AMV). The nowcasts include turbulence altitude as determined by an adapted height assignment scheme presented here. The case studies illustrate that the novel approach for satellite-based turbulence nowcasting is a supplement to the NWP models.

**Keywords:** aviation turbulence; eddy dissipation parameter; nowcasting; atmospheric motion vector; height assignment

## 1. Introduction

Almost 75% of weather-related flight incidents occur in areas of turbulence [1] or atmospheric areas of intense wind fluctuation. Turbulence is difficult to forecast because the phenomenon is produced by highly energy-intensive flow patterns, the lower spatial scale of which is limited only by the dimensions of the aircraft. Such small-scale structures cannot be adequately resolved on the computational meshes of current numerical weather prediction (NWP) models. Many of these turbulence phenomena, such as clear-air turbulence (CAT), mountain wave turbulence (MWT), convectively induced turbulence (CIT), or in-cloud turbulence (ICT) are characterized in diverse studies using high-resolution measurements or direct numerical simulations (see [1] for a detailed introduction). As these phenomena were not comprehensively considered directly in NWP, a number of turbulence indices were calculated in a postprocessing step diagnostically from the available model fields [1].

Over the last 15 years, Deutscher Wetterdienst (DWD) has developed and operationalized prognostic procedures, based on a universal turbulence parameter (the Eddy dissipation parameter, EDP) by parameterizing various subscale phenomena in the form of an extended turbulence model. The turbulence scheme used in the COSMO (Consortium of Small-scale Modelling) and ICON (Icosahedral Nonhydrostatic) models implements the prognostic equation for turbulent kinetic energy described in [2]. A series of important extensions to this equation for the time tendency of TKE are explained using a scale-separation approach [3,4]. One of the most important of the three additional source terms in the equation for the temporal change of the TKE describes horizontal shear eddies and is similar in some respects to the familiar Ellrod Index of vertical wind shear multiplied by the

difference from deformation and divergence of the horizontal wind. The expression is, therefore, composed of combinations of local derivatives of the wind components. Horizontal velocities can be taken from NWP forecasts. However, for short-term forecast, times up to 3–5 h, optical flow methods are able to provide accurate atmospheric motion fields [5] or horizontal velocities.

In this work, the optical flow method TV-L1 [6] is applied to satellite images for the estimation of atmospheric motion vectors (AMV). This opens the door for turbulence nowcasting because AMV provide the needed information for the calculation of the shear term of the TKE equation. Nowcasting procedures with optical flow are already used operationally at DWD for various weather hazards, such as convection [7,8], lightning, winter weather, precipitation, and convective initiation [9]. The use and the transfer of this technique to turbulence is described in the next chapters. The description of EDP calculation as a component of the turbulence scheme in the current NWP model ICON is given in Section 2, which is followed by Section 3 explaining the procedure for transferring the turbulence scheme to nowcasting with the final EDP turbulence product derived from satellite imagery. Section 4 continues with the approach of determination of the turbulence top height, and in Section 5, we look at some cases of real turbulent events. Finally, we present conclusions in Section 6 and an outlook.

## 2. Physical Basics Turbulence Scheme in the Current NWP Model ICON

The EDP is determined based on a deterministic forecast of the Eddy Dissipation Rate (EDR).

$$EDP = EDR^{1/3} \tag{1}$$

This physical quantity plays a key role in the parametrization of turbulence and agrees with Kolmogorov's theory of the spectrum of the TKE which can be expressed as

$$EDR = (2TKE)^{3/2}/\alpha\ell, \tag{2}$$
$$\alpha\text{ -Dissipation constant, } \ell\text{ - turbulence length scale}$$

Equation (2) is valid for inertial subrange spectra up to a maximum, isotropic, and turbulent wavelength. This assumption is included in the well-known closure model developed by [10] at level 2.5, according to the classification of Mellor and Yamada. The EDP was often far too weak in case studies of aircraft turbulence from this classic concept with the Blackadar profile for the turbulent length scales [10], which is specifically designed for the boundary layer (PBL). The use of various approaches for the length scales $\ell$ (e.g., stability dependent or as a fraction of the model grid distance) did not result in any improvement in the forecasting of areas of turbulence above the PBL [3]. It was almost impossible to reproduce CAT, CIT, ICT, and MWT, a more accurate description of the phenomena was only possible with the extended concept of scale separation [2,4]. This concept takes into account larger scale, nonturbulent subscale processes by replacing the formal filtering of model equations with a filter cascade for the closure of the equations. A filter for the horizontal separation scale is first applied. This simply filters out the turbulent part of the subrange spectra so that the TKE equation is valid [10]. Application of a second filter Dh, adapted to the horizontal numeric mesh resolution, produces additional TKE source terms. These scale transfer terms in the equation for the temporal change of the TKE (Figure 1) are subgrid scale (SGS), horizontal shear eddies (SHS), wake vortices by SGS orography and breaking gravity waves (SOW), and SGS convective vertical wind (SCV).

See [11,12] for more details about the derivation of the TKE components. In this work, we focus on the horizontal shear eddies (SHS), which are defined [11,12] by Equation (3).

$$SHS = c_1 \, Dh^2 \, (S/N)^{8/3} \, (\sqrt{((1.6268 DIV)^2 + DEF^2)} - 1.6268 \, DIV)^3 \tag{3}$$

Here, $c_1$ and c in Equation (4) are constants, Dh is the horizontal mesh size, DEF is the deformation, DIV is the divergence of the horizontal wind field, S is the vertical wind shear, and N is the buoyancy frequency (the ratio $(N/S)^2$ is the well-known Richardson number Ri). Using Ri as correction function is

important for the NWP model ICON, because the generation of vortices by horizontal shear is greater with weakly stable stratification than with very stable stratification. All additional source terms have the unit $m^2/s^3$ and are thus interpreted physically as a gradient flow. With Equation (2) results, the EDP (unit $m^{2'3}/s$) is only formed from the shear term according to [12].

$$EDP = c^3 SHS \qquad (4)$$

The main issue of this work is to develop a nowcasting method for the calculation of EDP based on satellite data. The high spatial and temporal resolution of the Meteosat Second Generation (MSG) weather satellite enables the calculation of atmospheric motion vectors (AMV); this refers to horizontal velocities, in different spatial scales relevant for aircraft turbulence, e.g., even small convective structures can be detected by the satellite-based nowcasting (like convective initiation) again [8,9].

Thus, we assume that AMVs are appropriate to simulate the wind fluctuations, which are responsible for the different kinds of aircraft turbulence. This hypothesis will be evaluated in Section 5.

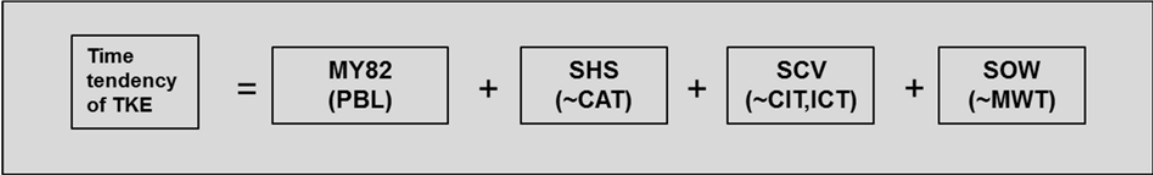

**Figure 1.** Advanced prognostic turbulence kinetic energy (TKE) equation, in principle and simplified form, MY82 according to [10] represents turbulence in the boundary layer (PBL). The notations in the brackets symbolized the corresponding parameterized phenomena.

## 3. Methodology EDP from Motion Vector and Satellite Data

This section further explains how the quantities in Equation (3) can be expressed by application of optical flow methods to satellite images. Optical flow (literally refers to the displacements of intensity patterns [13]) methods are well established in image processing. An overview of different optical flow methods and their applications can be found in [13–16]. In this work, the optical flow method TV-L1 by Zach et al. [6], implemented in OpenCV [5], is applied. The TV-L1 optical flow method was chosen based on its good performance for radar and satellite nowcasting at DWD, both in research studies and operational applications. It permits obtaining robust and discontinuity preserving solutions for optical flow with highly efficient implementations [6].

TV-L1 is already used at DWD for nowcasting radar echoes and cloud albedo [17]. TV-L1 outperforms Farneback and NWC-SAF AMVs [17] in the abovementioned application fields. However, for turbulence nowcasting, modifications and optimizations are needed. The calculation process in deriving turbulence nowcasting, based on TV-L1, involves five main steps:

1. Estimating the optical flow (AMV) for 6.2 and 7.3 µm MSG satellite water vapor channels from consecutive images (see Table 1).
2. Determining the wind components (sx, sy) by a transformation of the AMV between the image coordinate system (pixel per seconds) and the geographical coordinate system (meter per seconds). The conversions result from the Equations (5) and (6). So, divergence and deformation can be computed in physical metrics (Equation (7)).
3. Using the brightness temperatures ($BT_{62}$, $BT_{73}$) of the 6.2 and 7.3 µm water vapor channels (WV62, WV73) for the determination of vertical wind shear, static stability, and EDP (Equation (8)–(11)).
4. Deriving the turbulence top height with the so-called $H_2O$-intercept method (explanation in Chapter 4).

5.   The derived AMV's are then applied to the "observed" EDP in order to extrapolate up to 3 h into the future. Then after 3 h, a deterministic forecast from NWP delivers better results than satellite-based nowcasting in general [5,7,8,17].

**Table 1.** TV-L1 parameters (left, https://docs.opencv.org/3.4/dc/d47/classcv_1_1DualTVL1OpticalFlow.html) and projection parameters (right, with blue colored background).

| Parameter | Value | Parameter | Value |
|-----------|-------|-----------|-------|
| Tau | 0.125 | Type | regular lat/lon (eqc) |
| Lambda | 0.15 | lat_ts | 50 |
| Teta | 0.3 | lat_0 | 50 |
| Epsilon | 0.01 | lon_0 | 10 |
| outerIterations | 60 | A | 6378137.0 |
| innerIterations | 20 | B | 6356752.3 |
| Gamma | 0 | Height | 1113 pixels |
| scalesNumber | 5 | Width | 1193 pixels |
| scaleStep | 0.5 | lower left corner (xy) | -2146643.682, -1669792.3619 |
| Warps | 5 | upper right corner (xy) | 1431095.788, 1669792.3619 |
| medianFiltering | 3 | | |

The velocity field obtained through the optical flow procedure is in unit pixel per seconds. For the conversion addressed in item 2, we assume that each pixel in regular latitudinal and longitudinal projection has adjacent sides of equal length when expressed in meters. Each pixel has indices j along latitudes and i along longitudes. Calculations are performed using the WGS84 ellipsoid. For simplicity, however, we give the equations for a sphere.

$$\Delta x_{ij} = r\Delta\lambda\cos\varphi_{ij},$$
$$\Delta y_{ij} = r\Delta\varphi,$$
$$Dh_{ij}^2 = \Delta x_{ij}\,\Delta y_{ij} \tag{5}$$

Here, r is the earth radius, $\Delta y_{ij}$ is the distance in meters along longitude, and $\Delta x_{ij}$ is the distance in meters along latitude. Lambda is the longitude in radians, phi is the latitude in radians, $\Delta\lambda$ is the longitude pixel spacing in radian, and $\Delta\varphi$ is the latitude $\varphi$ pixel spacing in radiant. With (5), we transform the wind components $u_{x_{ij}}$, $u_{y_{ij}}$ from pixel/s to $s_{x_{ij}}$, $s_{y_{ij}}$ in m/s.

$$s_{x_{ij}} = u_{x_{ij}}\Delta x_{ij},$$
$$s_{y_{ij}} = u_{y_{ij}}\Delta y_{ij} \tag{6}$$

Deformation and divergence (unit 1/s) in Equation (3) are formed with the AMV from WV62 channel utilizing Equations (5) and (6), and it follows the expressions (7). This is motivated by the normalized weighting functions of Meteosat Second Generation satellite for clear sky [8]. The radiation intensity in the 6.2 µm band is very sensitive to differences in humidity of middle and high moist layers and, that is why, appropriate for upper level diagnostic (aircraft turbulence at cruising altitude).

$$DEF_{ij} = \sqrt{((\Delta s_{x_{ij}}/\Delta y_{ij} + \Delta s_{y_{ij}}/\Delta x_{ij})^2 + (\Delta s_{x_{ij}}/\Delta x_{ij} - \Delta s_{y_{ij}}/\Delta y_{ij})^2)},$$
$$DIV_{ij} = \Delta s_{x_{ij}}/\Delta x_{ij} + \Delta s_{y_{ij}}/\Delta y_{ij} \tag{7}$$

It is a serious challenge to gain information about the vertical structure of the atmosphere from satellite imagers. However, [9] showed successfully that vertical movement could be indirectly deduced from the satellite observations of the WV73 and WV62 channel. A vertical distance is required to determine the vertical wind shear of the horizontal wind of the AMV from WV62 and WV73 channel according to Equation (3). For this purpose, the gradient of the BT from 6.2 to 7.3 µm is transformed

with the commonly used standard atmosphere lapse rate of 0.0065 K/m for moist adiabatic processes, leading to the definition of the vertical distance d as

$$d = abs\ (BT_{62} - BT_{73}) \times 153.85 \tag{8}$$

The vertical wind shear with the unit 1/s is defined as:

$$S = \sqrt{((sx_{62} - sx_{73})/d)^2 + ((sy_{62} - sy_{73})/d)^2} \tag{9}$$

Brunt–Vaisala parameter N in Equation (3) requires the average temperature of the layer Tm.

$$Tm = (BT_{62} + BT_{73}) \times \frac{1}{2} \tag{10}$$

$$N^2 = (g/Tm)\cdot((BT_{62} - BT_{73})/d + g/cp), \\ \text{g-gravitational acceleration, cp-1005.7 J/kkg} \tag{11}$$

The Ri-dependent shear term (3) and the EDP (4) can thus be determined completely.

## 4. Methodology Turbulence Top Height TTH

The issue of determining the TTH is closely linked to AMV height assignment. See [18] for a detailed introduction. Various methods for different cloud types have been introduced: the carbon dioxide ($CO_2$)-slicing algorithm, the water vapor ($H_2O$) intercept method, the cloud-based technique, or the infrared-window approach. Both the $CO_2$-slicing method and the $H_2O$-intercept method were suggested for upper-level semitransparent clouds, utilizing the characteristics where differences between the measured radiance and the clear-sky radiance are greater than the instrument noise. In our work, the $H_2O$-intercept method in [19] is used. This method examines the relationship between clusters of clear and cloudy pixel values in water vapor-infrared window brightness temperature space, because radiances from a single cloud deck for two spectral bands vary linearly with cloud fraction within a pixel. The line connecting the clusters is compared to radiative transfer calculations for different cloud pressure levels. The intersection of the two lines gives the cloud height [19]. The radiative transfer calculation is typically performed for different standard atmospheres (subarctic winter, subarctic summer, midlatitude winter, midlatitude summer, and tropical). For our TTH approach, we supplement and validate the theoretical curve with actual radiation measurements, resulting from the brightness temperatures from satellite measurements for opaque clouds fulfilling the blackbody assumption. Under that condition, the BT from infrared (IR) 10.8 μm ($BT_{108}$) is equal to the physical temperature of the cloud top, which can be measured by radiosonde balloons (TEMP). A hundred radiosonde ascents were evaluated for each of the five atmospheres according to the following criteria: in an area with optically thick clouds, the difference of the brightness temperature of the WV channels [$BT_{62}$-$BT_{73}$] $> -1$ applies [8]. The radiosonde station lies in that area, taking into account the wind drift. In the high-range $BT_{108}$ of the TEMP-profile, the layer cloud analysis is applicable, in consideration of the relative humidity over ice (above dry below widespread moist). For example, at an assumed typical temperature of $-40$ °C at the 200 hPa level, a relative humidity (based on liquid water) of 70% would already be more than 100% relative humidity in relation to ice. The analysis leads to a data set containing measured values $BT_{62}$, $BT_{108}$, and height in hPa or flight level from TEMP-profile (Figure 2).

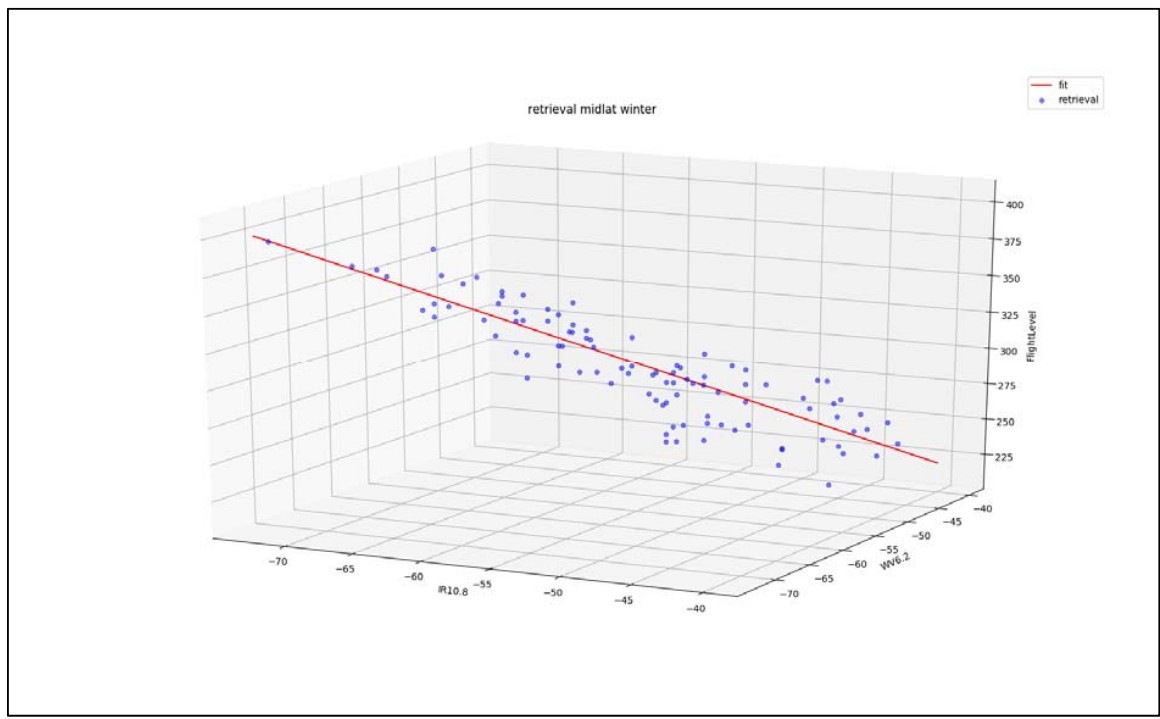

**Figure 2.** Brightness temperature (BT) infrared (IR) 10.8 μm, water vapor (WV) 6.2 μm with the unit °C, flight level with the unit hundreds of feet for midlatitude winter (blue), and best fit with multiple linear regression (MLR)-ordinary least squares (OLS).

To obtain an appropriate functional relationship between the data triples, multiple linear regression (MLR) is applied. The basic assumption is that the pressure altitude flight level (FL) depends on the BT's from IR 10.8 and WV 6.2 μm. This assumption enables us to fit a single line into the three-dimensional hyperplane using Equation (12). As fitting methods for the calculation of the regression coefficients are applied an ordinary least squares (OLS) regression and "Statsmodels," a Python module that provide classes and functions for the estimation of many different statistical models, as well as for conducting statistical tests is developed, and we have received the relationship

$$FL = r_0 + r_1 \cdot BT_{108} + r_2 \cdot BT_{62} \tag{12}$$

Here, $r_0$, $r_1$, and $r_2$ are the regression coefficients.

A calculation formula for the flight level according to $H_2O$-intercept method can be derived using linear algebra and analytical geometry, by identification of the intersection of a surface with a line.

In Section 3, the calculation of the EDP has been explained. For aviation applications dealing with "weather to cockpit," the information contained in the raster data has to be transferred to polygons in order to reduce the data amount and to generate a practical output format for distribution. Areas are thus described by means of vectors, and not raster data. The polygons enclose all pixels, whose EDP is greater than the threshold value for moderate turbulence taken from the NWP. The maxima and minima of the BT of the WV 6.2 μm ($WV_1$, $WV_2$) and IR 10.8 μm ($IR_1$, $IR_2$) channels are determined in the areas enclosed by each polygon. The differences between minimum and maximum should usually be large enough, so that the method works (see (13) $D_1$, $D_2$ not close to zero). Semitransparent clouds or subpixel clouds are often the best tracers for estimating AMV and, consequently, EDP. Since they show radiance gradients, that can easily be tracked [20]. The two pairs of values (minimum and maximum $IR_1$, $IR_2$ and $WV_1$, $WV_2$) can be used to form a straight-line equation at the bottom (FL = 0) and at the greatest possible height (FL = 500). Both linear equations constitute the equation for the plane with two direction and one position vectors. Now the intersection between the plane and the red line in Figure 2 can be identified. The cross-product of the direction vectors is formed for the

coordinate form of the plane equation. Now the line equation can be inserted into the plane equation and reorganized to flight level.

$$FL = 221 + t \cdot 180,$$
$$t = -((IR_1 \cdot D_1 - WV_1 \cdot D_2)/(D_1 - D_2) + 40)/33, \quad D_1 = 500(WV_2 - WV_1),$$
$$D_2 = 500(IR_2 - IR_1) \tag{13}$$

## 5. Results Case Studies of Strong Aircraft Turbulence

The evaluation of extreme turbulence events (values beyond 75 studies are documented for EDP from NWP) is a method for verifying new forecast procedures. The last two, most recent studies are supplemented by turbulence nowcasting and presented here. The current cases were selected because data availability was the most appropriate.

### 5.1. September 9, 2019: Airbus A319

On 9 September 2019, a weather front ahead of an altitude trough was located over east Europe with a surface depression near Bohemia. Corresponding with the shear zone of a jet, severe turbulence was expected in the area of East Saxony because several causes for aircraft turbulence (Figure 1) were apparent. Notably, the communication between airline, Federal Accident Investigation Board (BFU), and DWD clarified a decisive contribution by convection. An Eurowings airline Airbus A319 from Italy to Berlin encountered turbulence causing injuries to 3 cabin crew and 5 passengers. The incident occurred at 14:30 UTC between FL260 and FL235 during descent according to the flight transponder. The NWP (see Chapter 2) calculations of the event are presented in Figure 3 (visualized with the DWD "all-purpose forecaster workstation" NinJo). This three-dimensional representation of the forecast of the ICON model is only intended to show the overall situation along the flight route. In this case, no severe turbulence was predicted below FL250 by the NWP. Figure 4 shows the results of the turbulence nowcasting method described in Chapters 3 and 4 together with the NWP (transparent) for comparison. For this analysis, the NWP model level with the pressure altitude closest to the incident and the TTH from the nowcasting was selected (400 hPa or FL235).

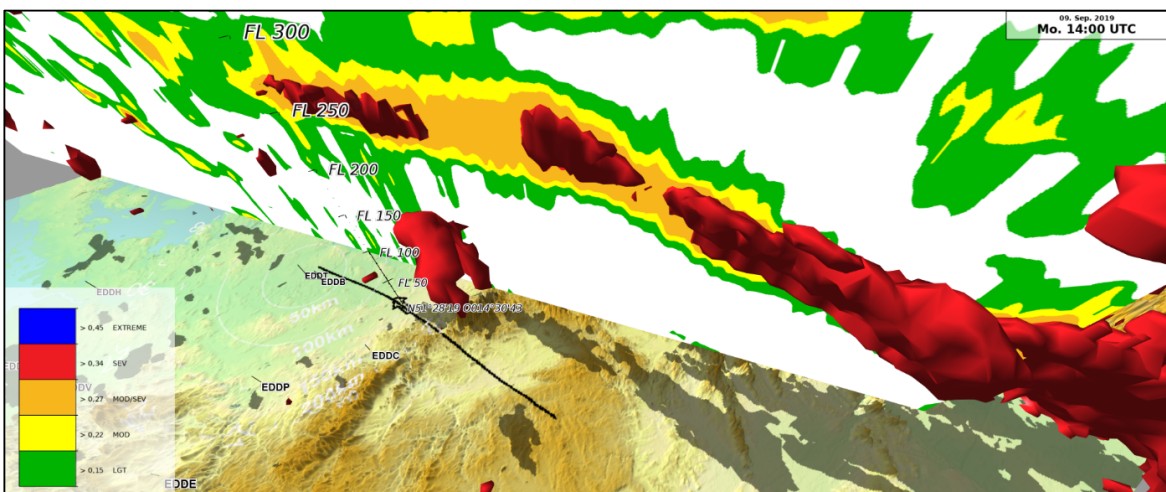

**Figure 3.** Numerical weather prediction (NWP) Icosahedral Nonhydrostatic (ICON)-Europe (EU) model Eddy dissipation parameter (EDP) forecast after 8 h for 14:00 UTC with severe (red), moderate (yellow), and light (green) turbulence. The arrow on the black line and the FL-lot-line symbolizes the area of incident. The black line on the ground corresponds to the A319 track with a length of 375 km between 14:15 and 14:45 UTC (airports EDDB-Berlin, EDDP-Leipzig, EDDC-Dresden). White: cross-section along flight path. The gray shadow is the projection of the red 3D turbulence isosurface on the ground.

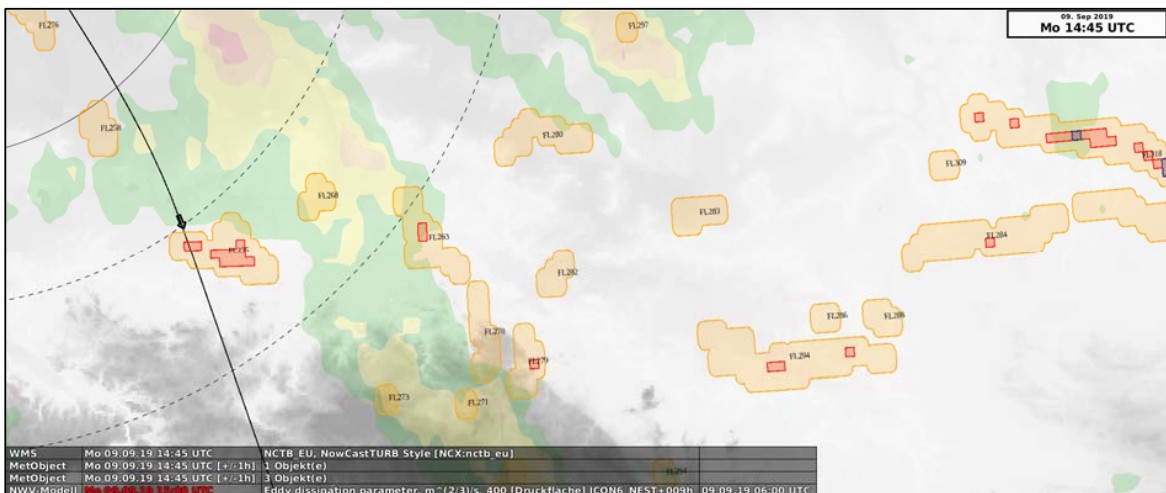

**Figure 4.** Nowcast-Turb 9 September 2019, 14:45 UTC; orange: moderate, red: severe, and blue: extreme turbulence polygons. The black line corresponds to the A319 track to Berlin. The arrow symbolizes the area of incident. There severe turbulence in FL235 was detected. Transparent in green, yellow, and red (color scale, see Figure 3): NWP ICON_EU model EDP forecast 15 UTC in FL235 (400 hPa). Circular dotted lines at a distance of 50 km, Berlin lies in the center of the circles.

Figure 4 shows that turbulence nowcasting can be a useful addition to the ICON NWP model in the short-term range, because severe turbulence (red polygon) correctly captures the flight trajectory while the model forecast (transparent red) is shifted about 50 km to the north. If this information was available, the pilot could, e.g., have set the seat belt sign or avoided the area altogether before the aircraft encountered the turbulence area.

## 5.2. March 2, 2020: Airbus A320

The second case is primarily related to the measured turbulence from a SWISS airline Airbus A320. It is unknown if the incident resulted in injuries or damage. It becomes a further important property of EDP a physical based atmospheric turbulence quantity, clearly derivable from weather prediction and aircraft measurements. A dedicated on-board software, developed at NCAR [21], was used to derive final estimates from the measurement records of vertical aircraft acceleration or speed, that are neither affected by properties of the individual aircraft or on flight maneuvers. On 2 March 2020, a frontal system over the central Mediterranean area with a depression near northern Italy generated a shear zone associated with a discontinuity area (upper front) near Montenegro too. Here are excerpts from the crew report of the flight: "During Cruise after about 1.20 h flight time with occasional light turbulence ( . . . ) there has been no turbulence for the last 10 min. Suddenly we encountered moderate to severe turbulence with some heavy bumps. ( . . . ) The first bumps made the trolleys in the rear galley leave the ground, fall to the side, and block the seats and the toilet door. ( . . . ). We informed ATC." The recorded measurement (turbulence determination according [21]) associated with this crew description was 0.42 $m^{2/3}$/s (corresponding to severe turbulence) at 21:56 UTC in FL350 at 42.59° N 18.57° E. The ICON NWP model calculations of the event are presented in Figure 5. Again, this three-dimensional representation only serves to illustrate the overall situation along the flight path. Figure 6 shows the results of the turbulence nowcasting method described in Chapters 3 and 4 together with the NWP (transparent) for comparison. For this analysis, the NWP model level with the pressure altitude closest to the measure and the TTH from the nowcasting was selected (240 hPa or FL350).

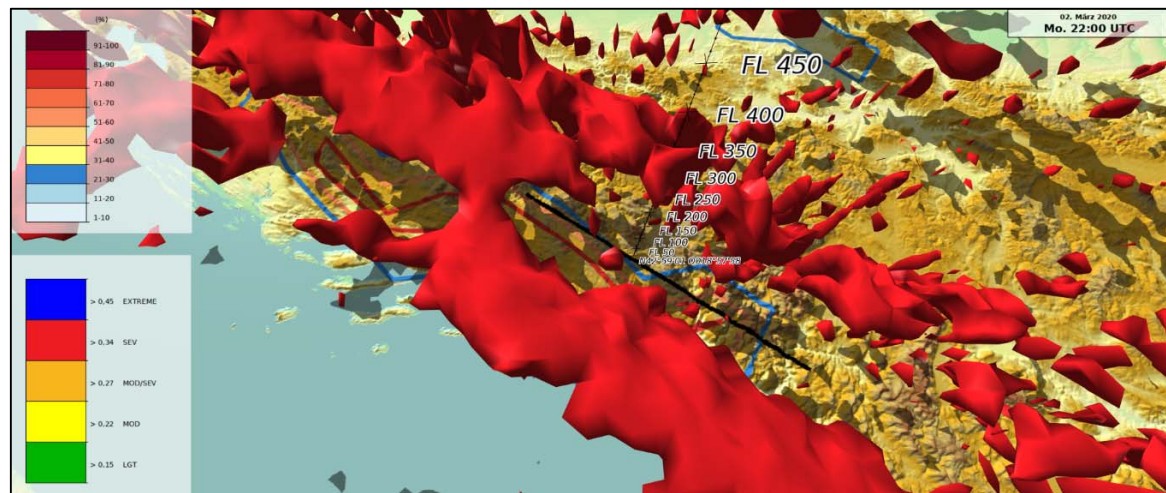

**Figure 5.** NWP ICON-EU model EDP forecast at 22:00 UTC on 2 March 2020 with severe (3D red isosurface) turbulence. The FL-lot-line marks the A320 spot of severe turbulence encounter with the base at N42.59° E18.57°. The colored isolines on the ground (2D for FL350) are from the ICON-EU-EPS ensemble prediction system with the EDP-probability of exceeding the value 0.34 (severe turbulence) from the 40 individual solutions. EDP values up to 0.51 m$^{2/3}$/s (equivalent to extreme turbulence, lower color scale, see Figure 3) and ICON-EPS EDP probability over 50% (upper color scale red) in FL350 was predicted. The black line on the ground corresponds to the A320 track with a length of 210 km between 21:50 and 22:05 UTC.

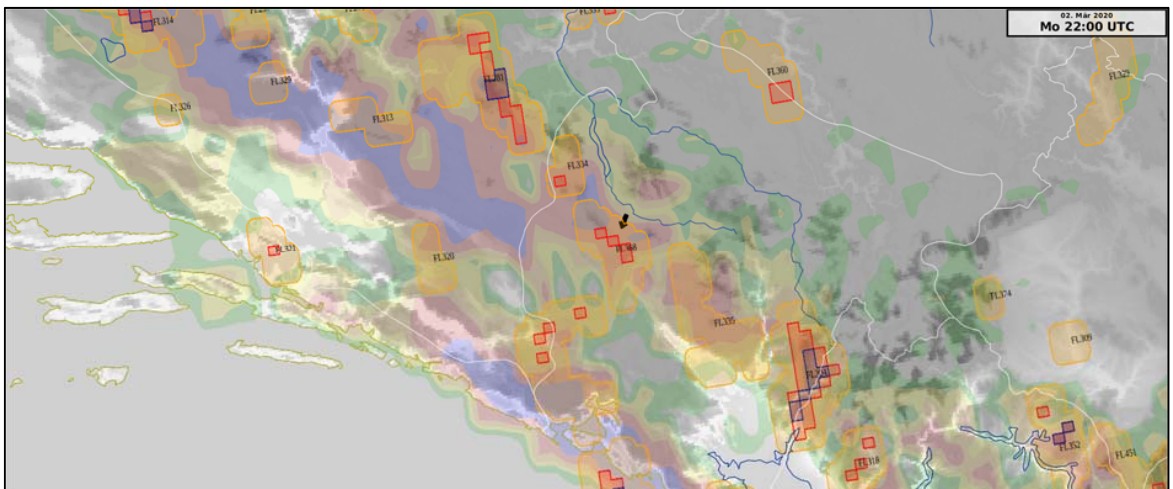

**Figure 6.** Nowcast-Turb 2 March 2020 22:00 UTC; orange: moderate, red: severe, blue: extreme turbulence polygons. The black arrow corresponds to the A320 measurement. Near the arrow, symbolizing the area of incident, severe turbulence in FL370 was detected. Overlaid is the transparent NWP 22:00 UTC forecast in FL350 of EDP in green (light), yellow (moderate), red (severe), and blue (extreme) with color scale, see Figure 5. The white line symbolizes the border of Montenegro.

Additionally, in this example, Figure 6 illustrates that the nowcasting can detect correctly severe turbulence that has occurred and predicted by the model ICON. The flight level data show relatively small deviations, compared to known errors of the traditional AMV height assignment (up to 50 hft [22]).

## 6. Conclusions

The advanced turbulence scheme as part of the scale-separation concept is used in the NWP model ICON and includes additional source terms in the existing prognostic TKE equation, in particular, the shear term. Based on this, the EDP can be derived from Meteosat Second Generation satellite water

vapor 6.2 and 7.3 μm and infrared 10.8 μm channel. Together with the adapted $H_2O$-intercept method for determining the turbulence top height, it is shown within case studies that the novel method for turbulence nowcasting provides information. In turn, this means that the resolution of the satellite data is high enough and that derived atmospheric motion vectors are appropriate to simulate the wind fluctuations, which are responsible for different kinds of aircraft turbulence. This needs to be underpinned by further case studies (in particular with MWT or CIT) and a comprehensive verification. The study provides clear indications that our nowcasting method assists the NWP in the short-term range, especially if convection is present, because the technique can detect turbulence and predict it for brief periods. The horizontal grid scale of current global model is indicated by the necessity to apply additional parameterization schemes, say for SGS-convection or wake-circulations generated by SGS-orography. The higher resolution-nested local model (ICON_EU) has been shown to be able to reproduce patterns of scale-related wind shear caused by mountains (MWT). This supports the thesis that the even higher resolution satellite data can also detect these movement patterns, as in the case of convection too [8,9]. The functionality of the so-called regularizer of the TV-L1 method might be not optimal to identify divergent and deforming features (Equation (3)) in satellite imagery. The regularizer discussed in [14,16] might provide the option to overcome this limitation and to improve the turbulence nowcasting. This needs to be further investigated in forthcoming works.

A future turbulence forecasting process will need to consist of several components including nowcasting and (ensemble-based) NWP. In this way, the advantages of both approaches can be sensibly combined and the weaknesses can be compensated with the aim of making a seamless prediction [7]. The actual benefit of the new method will be determined by a comprehensive verification during further development with a specification of the relevant probability of detection, false alarm rate, and receiver-operating characteristic [11,12]. The forecasting performance of nowcasting as a function of the time step (1, 3, and 5 h) will also be investigated. This clarifies the urgent need to increase the availability of those measurements over Europe significantly, and is expected to happen this year after the algorithm according [21] is implemented on Lufthansa Group aircrafts. With corresponding results, an expansion of nowcasting from the European (MSG) to the global domain is conceivable afterwards. Finally, the available operational display of the EDP for forecasters, air traffic controllers, airline operators, or in the electronic flight bag (EFB as part of weather to cockpit) for the pilots should be revised. It is considered that the combination of all turbulence information (EDP deterministic, EDP probabilistic, nowcasting, and measurement) in one product ("weighted EDP") using fuzzy logic is the last step to facilitate the pilots' risk assessment.

**Author Contributions:** A.B. developed the method and performed the case studies supported by S.H. and R.M. M.J. managed the project. All authors have read and agreed to the published version of the manuscript.

**Acknowledgments:** We thank the founder of the ICON turbulence scheme Matthias Raschendorfer from DWD for his long-term support and advice. Many thanks also to SWISS and DLH Airline (LH-group) for cooperation and providing the data and Swiss SkyLab Foundation and the Swiss Federal Office for Civil Aviation (BAZL) for implementing the measurement algorithm on airplanes.

**Conflicts of Interest:** The authors declare no conflicts of interest.

## Abbreviations

The following abbreviations are used in this manuscript:

| | |
|---|---|
| AMV | atmospheric motion vector |
| BT | brightness temperature |
| CAT | clear-air turbulence |
| ICT | in-cloud turbulence |
| CIT | convectively induced turbulence |
| DLH | German Airline "Deutsche Lufthansa" |
| DWD | Deutscher Wetterdienst |
| EDP | Eddy Dissipation Parameter |
| EPS | ensemble prediction System |

| EU | Europe (domain) |
| FL | flight level |
| ICON | NWP model of Deutscher Wetterdienst (Icosahedral Nonhydrostatic) |
| IR | infrared (channel, e.g., 10.8 µm band) |
| MSG | Meteosat Second Generation |
| Meteosat | meteorological satellite |
| MWT | mountain wave turbulence |
| NWP | numerical weather prediction |
| SEVIRI | spinning enhanced visible and infrared imager |
| SGS | subgrid scale |
| SWISS | synonym for Switzerland |
| TEMP | synonym for the measurement as well as evaluation of the data collected via radiosonde ascent |
| TKE | turbulence kinetic energy |
| TTH | turbulence top height |
| WGS84 | World Geodetic System as a reference coordinate system |
| WV | water vapor (channel, e.g., 6.2 or 7.3 µm band) |

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
