# Peer review of "A Novel Approach for Satellite-Based Turbulence Nowcasting for Aviation"

_remotesensing, doi:10.3390/rs12142255_

Round 1

Reviewer 1 Report

See attached file.

Author Response

Dear Reviewer 1

In the "Cover letter editor" we have briefly explained why we have not yet responded to your major revisions. We will make up for this next week. However, we cannot consider your remarks in the article anymore.

Reviewer 2 Report

This seems to be a useful piece of work which will be highly relevant to aviation safety.  In principle, it merits publication.

I have some concerns about Section 3. From ll82-83, we see that S is the vertical wind shear and N the 'buoyancy'.  Then Ri = N/S.  But conventionally N denotes the Brunt-Vaisala frequency and then Ri = (N/S)2.  Later on (Equation 8) we see that S is indeed the vertical wind shear, and (Equation 10), but N is the square of the Brunt-Vaisala frequency.  I hope that their program does then indeed calculate Ri as N/S2.  It would be preferable to use the conventional notation for the Brunt-Vaisala frequency.  I note also that they use the standard atmosphere for the temperature gradient in calculating N, but then say (ll144-146) that stability is also important in calculating the turbulence generation.

BTW I assume that Equation (9) should contain a plus sign.

I found Figure 3 quite hard to follow.  Would it be possible to tie the position of the aircraft along its track to a position on the vertical section?  Likewise Figure 5 was difficult, but then I suppose the authors must accept the system output that they are given.

There are a few minor grammatical/typographical glitches, e.g. l275 'which are' instead of 'who are'.  Some sub-editing might be helpful.

Reviewer 3 Report

This paper presents a method for nowcasting and short-term forecasting (up to three hours) of clear-air turbulence based on analysis of satellite-derived multilevel wind fields. The wind fields are used to estimate a Richardson number and thereby diagnose high-turbulence areas (assuming a typical static stability). The future locations of turbulence are then projected by advection using the retrieved wind. The method is applied to two case studies where turbulence was reported during commercial flights, and appears more successful in both cases than currently available forecasts from met services.

The results are interesting and promising and I think this paper should be published with revisions. The main problems that need addressing in revision are:

  1. The writing could stand improvement. Most importantly, it is difficult to discern from the abstract or conclusions what the authors did, and what its value is. Part of the problem stems from some inappropriate terminology, for example the authors say they are developing a “parameterization” and imply this is to be used in a numerical forecast model, discussing a lot about these models, when in fact they are proposing a separate prediction method using only satellite data which happens to use equations for turbulence prediction that are currently used in numerical models. As far as I can tell they are not developing a model paramaterization.

    Another problem is that the English is often hard to follow or has grammatical errors (some detailed below).

    A third problem is that many acronyms are defined and most don’t seem necessary. This is an impediment for readers, they are hard to keep track of. I personally dislike acronyms for numerical quantities, is better to use symbols (Greek or arabic letters with subscripts as necessary).  Acronyms are better used for concepts or things, such as CAT etc.
  2. The authors do not say how they chose the two cases shown.  This is important since, for example, they could have looked at dozens of cases and only shown the two that worked.

  3. Some symbols are not defined, and there is a general problem of the authors not giving units where necessary, or fully defining some quantities (see some examples below).

  4. The paper does not say much about previous efforts to forecast clear-air turbulence (see for example doi: 10.1007/s10994-013-5346-7 and I would expect there are other papers).

Detailed comments to the authors.

59: You seem to be using a single quote mark ‘ in superscripts for fractions, where you should have a /

63: can you explain what “at level 2.5” means

64-6: sentence unclear, please rewrite

68-70: run-on sentence

79: caption does not make sense

82: there is no ‘c’ in the equation. Also, D_h is not defined.  DIV and DEF are the divergence/deformation of what?  the horizontal wind field?

96: I believe “cloud (or atmospheric) motion vectors” is the usual terminology for this, or is that something different?  Optical flow is a strange term since we are not talking about the flow of photons, and are not using optical wavelengths. Use whatever term you prefer but please mention on first use how it is related to other terms in use.

117: here and elsewhere the manuscript states “buoyancy” when it should say “buoyancy frequency” (or Brunt-Vaisala parameter or static stability). “Buoyancy” on its own normally refers to a horizontal difference in air density.

119-121: What is the basis for this statement?  The manuscript says later that a proper evaluation of skill is yet to be done and only two case studies are shown.

125 what is the WGS84 ellipsoid?  Need a definition or citation.

140 "Standard" lapse rate is not exactly moist adiabatic, it is basically empirical (and not always accurate). I think a key point to make is that in the troposphere above the PBL the lapse rate is much less variable than the wind shear, so if you know the wind shear you can get a rough estimate of the Richardson number. BTW the stability will increase substantially closer to the tropopause (6.5 K/km will not work any more)—what are the limits to the altitude range where this method can be applied?

141: equation needs to give units

143: Buoyancy *frequency* N in ….  Also, the equation for N has a sign mistake unless g is defined negative (in which case the sign needs to be clarified)

174: A scatter plot really doesn’t work in three dimensions. What is this plot meant to convey?  Also the axis labels need units.

177: do we need the acronym FL?  Is this a pressure or an altitude (in which case I’d suggest p or z respectively with some subscript to indicate flight??

189: remove comma

193-4: logic of this statement escapes me (seems like cause and effect are being switched), and it has grammatical problems.

196: 500 what?

216: Legend text is illegibly small and airport names are also very hard to find/read. What is the grey shading, is this “shadows” showing the surface footprint of the turbulence features rendered aloft?

222: why are dotted circles shown, what is at the centre of these?

228-30: logic confusing. I think what you mean is that the system correctly diagnosed severe turbulence near the arrow, where it was encountered by the aircraft, whereas the weather service forecast did not predict any problem in this area.

233: why is SWISS in all caps?  Is this another acronym for something?

246: What is meant by “the recorded measurement”, is this your diagnostic from remote sensing or a measurement from the aircraft and what exactly is the quantity?

259: why isn’t the flight track included in this figure?

Reviewer 4 Report

Dear authors, 

please, find attached my repport. You need to change many aspects related to the format of the manuscript.

Best regards

Round 2

Reviewer 1 Report

The provided response did not address multiple concerns (Major and Minor) I raised in review rounds 2 and 3, so my recommendation will not change with this round.  I am frankly frustrated that the authors believe we are talking past one another and trying to circumvent my comments with this reply rather than simply addressing the problems I raise within the manuscript.  If the authors do not understand the issues I raised within review rounds 2 and 3, then it is my opinion that they have not performed sufficient background research and experimentation for the problem at hand, and are not prepared to publish this work.

It would be an improvement to mention the Heas et al. and Corpetti et al. studies within the manuscript, though the authors should discuss the limitations with the TV-L1 approach as well, and how the limitations may impact the turbulence parameterization performed here.  This way, the readers will have a healthy understanding of the approach used (how it penalizes divergence and deformation within the derived flow solutions), and where improvements can be made in the future.  With the limitations stated clearly somewhere in the article, I feel as if my Major Comment 1 in Review Round 2 would be addressed.  However, there are still major comments 2 and 3 from review round 2 (regarding the conclusions stated in this article and the cloud-edge issue), Major Comment 1 from review round 3 (regarding the authors' claims of 1-5 hour nowcasting), and a few minor comments (e.g. adding radiometric resolution and scaling to Table 1) that still must be addressed within the manuscript prior to publishing, regardless of whether or not it is submitted as a tech-memo.

Author Response

Dear Reviewer 1,

In order to do justice to your major comment, we have added in the conclusions,

The functionality of the so-called regularizer of the TV-L1 method might be not optimal to identify divergent and deforming features (equation (2)) in satellite imagery. The regularizer discussed in [3,9] might provide the option to overcome this limitation and to improve the turbulence nowcasting. This needs to be further investigated in forthcoming works.

Many thanks for your help and support

The authors

Reviewer 4 Report

Dears authors,

congratulations for the notable improvement of the manuscript

It is nearly to the acceptation, however, in my opinion, some minor issues should be solved:

  • the equations must be clearly exposed, the equation themselves, but also the text explaining them
  • there are some notations that present different letters, depending on the part of the doc. Please, unify (one example is IR_10.8 or IR10.8)

Best regards

Author Response

Dear Reviewer 4,

Many thanks for the appreciative words, this has motivated us further to implement their last hints. We have adapted the notations in the text and in the formulas. That was once again very important. Therefore, we want to thank you explicitly for the constructive cooperation.

The authors

This manuscript is a resubmission of an earlier submission. The following is a list of the peer review reports and author responses from that submission.

Round 1

Reviewer 1 Report

I found the article well written and the results meaningful. However, I would recommend some minor revisions:

  1. Change the reference ([4]) to the advanced prognostic TKE equation (Fig.1), since it has not yet been published. Alternatively, improve the relative theoretical treatment;

  2. Regarding the explicit description of your algorithm (lines 110-120).
    It is clear to me how you derive the components of the (instantaneous) EDP (according to eqs. 2 and 3) from satellite data, but how do you project it into the future? If you compute EDP from SHS (Eq. 3) you need a prognostic equation for it. The other alternative is that you compute EDP from TKE (Eq. 1) and use its prognostic equation (reduced to SHS and MY82 only). You should clarify this point;

  3. I would suggest to enrich section 5, by discussing differences and similarities between your method and NWPs more exhaustively.

Reviewer 2 Report

See attached file.

Round 2

Reviewer 2 Report

See attached file.
